# An Insight into the Exploration of Antibiotic Resistance Genes in Calorie Restricted Diet Fed Mice

**DOI:** 10.3390/nu15143198

**Published:** 2023-07-19

**Authors:** Xiuqin Fan, Yuanyuan Lu, Yunfeng Zhao, Hongjian Miao, Kemin Qi, Rui Wang

**Affiliations:** 1Laboratory of Nutrition and Development, Key Laboratory of Major Diseases in Children, Ministry of Education, Beijing Pediatric Research Institute, Beijing Children’s Hospital, Capital Medical University, National Center for Children’s Health, Beijing 100045, China; qincaifan@126.com (X.F.); qikemin@bch.com.cn (K.Q.); 2Department of Children’s Health Care Center, Beijing Children’s Hospital, Capital Medical University, National Center for Children’s Health, Beijing 100045, China; luyuan1988220@163.com; 3NHC Key Laboratory of Food Safety Risk Assessment, Chinese Academy of Medical Sciences Research Unit (2019RU014), China National Center for Food Safety Risk Assessment, Beijing 100021, China; zhaoyf@cfsa.net.cn

**Keywords:** antibiotic resistance genes, mobile genetic elements, calorie restricted diet, gut microbiota

## Abstract

Antibiotic resistance genes (ARGs) threaten the success of modern drugs against multidrug resistant infections. ARGs can be transferred to opportunistic pathogens by horizontal gene transfer (HGT). Many studies have investigated the characteristics of ARGs in various chemical stressors. Studies on the effects of dietary nutrition and dietary patterns on ARGs are rare. The study first demonstrated the effect of calorie restricted (CR) diet on the ARGs and mobile genetic elements (MGEs) in mouse feces and explored their relationship with gut microbiota and their functions. The results showed that the abundance of the total ARGs in mouse feces of the CR group increased, especially tetracycline ARGs (*tet*W-01). The abundance of the MLSB ARGs (*erm*B) decreased evidently in mouse feces of the CR group. In addition, the total abundance of MGEs decreased evidently in the CR group, especially *tnp*A-03. In the meantime, the abundance of *Lactobacillus* and *Bifidobacterium* in mouse feces of the CR group increased remarkably. The Spearman correlation analysis between gut microbiota and ARGs showed that several probiotics were significantly positively correlated with ARGs (*tet*W-01), which might be the main contribution to the increase in ARGs of the CR group.

## 1. Introduction

The global rise of ARGs threatens the success and sustainability of modern drugs against deadly multidrug resistant infections [1,2]. A recent study showed that 1.27 million people died directly and 4.95 million died indirectly from antimicrobial resistance (AMR) in 2019 [3], with projections reaching 10 million deaths per year by 2050 [4]. Thus, the World Health Organization (WHO) suggests an urgent need for globally coordinated response measures and an improvement in our basic understanding of ARGs [5,6]. It is well accepted that human gut microbiota are important reservoirs of ARGs [7]. Most gut microbiota have a symbiotic relationship with their host, but also contain opportunistic pathogens, including those belonging to the families *Enterobacteriaceae*, particularly *Escherichia coli.* and *Klebsiella pneumonia*, and *Enterococcaceae*, most notably *Enterococcus faecalis* and *Enterococcus faecium* [4,8].

ARGs can not only be exchanged among the commensal microbiota but can also be transferred to opportunistic pathogens, which is challenging life-saving antibiotic therapies [9]. The acquisition of ARGs from other strains of the same or different species occurs in a process termed HGT [10], which includes conjugation, transformation, and transduction [11]. MGEs, such as plasmids, insertion sequences (ISs), integrons, transposons (*tnp*), and phages have specific structures and ability to capture genes by a site-specific recombination system [12,13]. Intrinsic and acquired resistance by mutation are thought to present a low risk of horizontal spread, but acquired resistance mediated by MGEs is thought to have the highest risk of horizontal spread [14]. In addition, MGEs are being studied as agents with their own interests and adaptations, and the interactions MGEs have with one another are recognized as having a powerful effect on the flow of traits between microbes [15].

The main mechanisms by which bacteria develop resistance to antibiotics are prevention of the antibiotic from reaching toxic levels in the cell, modification of the antibiotic target, and modification or degradation of the antibiotics themselves [9,16]. At present, many studies have investigated the characteristics of ARGs in various chemical stressors, including antibiotics, heavy metals, nanomaterials, and disinfectants which serve as drivers of the ARG development [7,8,9,10,11,12,13,14,15,16,17,18,19,20,21]. However, studies on the effects of dietary nutrition and dietary patterns on ARGs are rare. Our previous research has proven that only obesity induced by a high-fat diet could accelerate the dissemination of ARGs in mouse gut in the absence of exogenous chemical selective pressures [22]. The potential mechanism was related to body weight and the changes in the mouse gut microbiota induced by body weight [22]. Therefore, the inherent risk levels from a given ARG could be small compared to the overall picture of risks for the spread of ARGs in both clinical practice and the environment; that is, the genomic contexts of a bacterial population matter more than the identity and quantity of ARGs themselves [23].

Of course, high-fat diet (HFD) induced obesity is a way to gain body weight, while a calorie restricted (CR) diet is a way to reduce body weight. Nowadays, CR is the only regimen known to extend the life span and health span in a spectrum of organisms, including mice and non-human primates. CR diet also serves as an effective approach to promote physical health, as well as remodel gut commensal microbiomes dramatically [24]. Reducing food consumption 25–60% without undernutrition extends the life span of rodents up to by 50% [25]. Additionally, CR conferred protective effects on long-term rehabilitation of stroke, partly through gut microbiota modulation, especially by the enrichment on *Bifidobacterium* [24]. Extreme calorie restriction induced intestinal stem cell dysfunction, resulting in impaired de novo generation of digestive epithelial mass, and these changes are also microbiota-dependent [26].

However, there is currently no literature reporting whether loss of body weight and modification of gut microbiota caused by calorie restricted diet could affect the dissemination of ARGs and MGEs. Since obesity induced by an HFD could accelerate the spread of ARGs in the mouse gut, is it possible that loss of body weight caused by a calorie restricted diet could affect the ARGs from the opposite side? Thus, we hypothesize that a calorie restricted diet might have an influence on the spread of ARGs. This study first demonstrated the variations of ARGs in mouse gut under the condition of a calorie restricted diet and then compared the difference in the ARGs in mouse gut between a high-fat diet and calorie restricted diet and further explored the potential factors affecting the diffusion of ARGs.

## 2. Materials and Methods

### 2.1. Animal Studies

Eight-week-old male C57BL/6J mice were purchased from the SPF (Beijing) Biotechnology Co., Ltd. (Beijing, China) and were housed at the animal facilities in the National Institute of Occupational Health and Poison Control, China CDC, under a 12 h dark–light cycle with constant temperature (23 °C) and free access to water and food.

CR is an eating pattern that involves reducing calorie intake (typically by 10–50% of ad libitum consumption) without malnutrition that extends the healthy lifespan and reduces incidence of chronic diseases [27]. Typically, CR diets feature a level of food that is 60–70% of what animals would eat ad libitum (AL) [24]. Thus, we chose the level of 65% reduction of total dietary consumption. Mice were housed in individual cages and fed AIN-93M diet (Beijing Huafukang Bioscience Co. Inc., Beijing, China) ad libitum until baseline food intake could be calculated. Then, mice were randomly divided into two groups (n = 10 per group): AL mice were fed AL and CR mice were fed 65% of their AL food intake. The body weight and food intake of mice were measured weekly. Fresh stool samples were collected at the end of the CR interventions and stored at −80 °C.

All animal procedures in this study were performed in accordance with the recommendations in the Guide for the Care and Use of Laboratory Animals of National Administration Regulations on Laboratory Animals of China. All experimental protocols were approved by the Committee on the Ethics of Animal Experiments of the National Institute of Occupational Health and Poison Control of China.

### 2.2. DNA Extraction and ARG Analyses

Approximately 200 mg fresh feces of mice from each group (n = 10) was applied for total genomic DNA extraction using a QIAamp DNA Stool Mini Kit (QIAGEN; Solon, OH, USA) according to the manufacturer’s instruction. The concentration and quality of the extracted DNA were determined by 1.5% (*w*/*v*) agarose gel electrophoresis and NanoDrop 1000 spectrophotometer analysis (NanoDrop; Wilmington, DE, USA). A total of 88 primer pairs were set for targeting 74 ARGs, 10 MGEs, 3 pathogenic genes, and the 16s rRNA gene and the details are shown in Appendix A. All high-throughput (HT) qPCR reactions were carried out by the Wafergen SmartChip Real-time PCR system as described by Wang et al. [22]. Cycle thresholds (CTs) were applied to calculate copy number of genes via copy number = 10^(30-CT)^/^(10/3)^, using the cutoff threshold of CT < 30. Relative abundance of detected genes was calculated by dividing by the copy number of the 16s rRNA gene.

The 74 ARGs detected contain major ARGs, including aminoglycoside (11), beta-lactamase (14), macrolide–lincosamide–streptogramin (mlsb) (10), multidrug (15), sulfonamide (1), tetracycline (12), vancomycin (7), and others (4), which covered 6 resistance mechanisms (antibiotic efflux, antibiotic inactivation, antibiotic target alteration, antibiotic target protection, antibiotic target replacement, and unknown).

### 2.3. Microbiota Community Analyses

DNA extracts (two DNA extracts from Section 2.2 mixed as one sample, n = 5) were fragmented to an average size of about 400 bp using Covaris M220 (Gene Company Limited, Shanghai, China) for paired-end library construction. A paired-end library was constructed using NEXTFLEX Rapid DNA-Seq (Bioo Scientific, Austin, TX, USA). Adapters containing the full complement of sequencing primer hybridization sites were ligated to the bluntend of fragments. Paired-end sequencing was performed on Illumina NovaSeq/HiseqXten(Illumina Inc., San Diego, CA, USA) at Majorbio Bio-Pharm Technology Co., Ltd. (Shanghai, China) using NovaSeq Reagent Kits/HiSeq X Reagent Kits according to the manufacturer’s instructions (www.illumina.com (accessed on 17 December 2022)). Sequence data associated with this project have been deposited in the NCBI Short Read Archive database (Accession Number: SRP443205). The data were analyzed on the free online platform of Majorbio Cloud Platform (www.majorbio.com). The paired-end Illumina reads were trimmed of adaptors, and low-quality reads (length < 50 bp) were removed by fastp [28] (https://github.com/OpenGene/fastp (accessed on 17 December 2022), version 0.20.0).

Reads were aligned to the mouse genome by BWA (http://bio-bwa.sourceforge.net (accessed on 17 December 2022), version 0.7.9a) and any hits associated with the reads and their mated reads were removed. Metagenomics data were assembled using MEGAHIT [29] (https://github.com/voutcn/megahit (accessed on 17 December 2022), version 1.1.2), which makes use of succinct de Bruijn graphs. Contigs with a length of 300 bp or more were selected as the final assembling result, and then the contigs were used for further gene prediction and annotation.

### 2.4. Data Analyses

Statistical calculations and data analysis were carried out by the SPSS 16.0 statistical software and Origin 9.0 (Origin Lab., Northampton, MA, USA). The Kolmogorov–Smirnov test was used to evaluate whether the data were normally distributed. We used the unpaired *t*-test for the normally distributed data and the Mann–Whitney U test/Wilcoxon rank-sum test for the non-normally distributed data to calculate the difference between the CR and the AL groups, where *p* < 0.05 was considered statistically significant. Principal component analysis (PCA) and redundancy analysis (RDA) were conducted using Canoco 5.0 to determine the correlations between ARGs and microbial community. The percentage of variation explained by each axis is shown in the figures, and the significance of the relationship was determined (*p* < 0.001) based on 499 permutations.

## 3. Results and Discussion

### 3.1. CR Mouse Model

To evaluate the influence of CR on the variations of ARGs, we reduced the calorie intake of mice to 65% (a 35% CR) of that fed ad libitum mice in early adulthood (10 weeks of age), and this dietary regimen was maintained until 28 weeks of age (Figure 1A,B). No difference existed in body weight at baseline between AL (25.1 ± 0.9 g) and CR (24.9 ± 0.9 g) mice. After 1 week of CR intervention, CR led to a reduction in body weight by 15.9%, while AL mice had an increase in body weight of 4.6%. Then, CR mice remained lean throughout the experimental period (Figure 1C). After 18 weeks, the body weight of AL fed mice continuously increased 6.6 ± 1.2 g with aging up to 31.7 ± 1.5 g, whereas CR mice lost weight and weighed 21.8 ± 0.6 g at the end of the study, 3.2 ± 1.0 g less than their initial weight (Figure 1C,D). The body weight changes in AL and CR mice are consistent with other previous studies [30,31].

### 3.2. Effect of CR on ARGs, MGEs, and Pathogenic Genes

According to the previous research [22], a total of 88 genes, including 16s rRNA genes, 74 ARGs, 10 MGEs, and 3 pathogenic genes, were selected and determined in the mouse fecal samples between the CR and AL groups. The abundance of total ARGs in the mouse feces of the CR groups was higher than that of the AL groups. Especially, the most abundant ARGs, tetracycline ARGs (*tet*W-01), increased 8.8 times in the CR groups compared to that of the AL groups (Figure 2A,D). Interestingly, the abundance of MLSB ARGs (*erm*B) in the mouse feces of the CR groups decreased significantly compared to that of the AL groups. However, *erm*B was the most significantly increased ARG in the high-fat diet induced obese mouse feces in the previous study [22], which exhibited an opposite trend to that in the CR groups. The relative abundance of sulfonamide resistance genes (*sul*2) increased significantly in the mouse feces of the CR groups compared with that of the AL groups (Figure 2J), which also exhibited an opposite trend to that in the high-fat diet induced obese mouse feces [22]. It might indicate that *erm*B and *sul*2 were body weight-associated genes in gut microbiota.

Additionally, the abundance of aminoglycoside (Figure 2F) and vancomycin (Figure 2I) resistance genes showed a decreased trend and the abundance of β-lactamase (Figure 2G), multidrug (Figure 2H), and other (Figure 2J) resistance genes showed an increased trend in the mouse feces of the CR groups compared with those of the AL groups. The RDA results (Figure 3A) showed that the ARGs clustered around the CR group (lower body weight) mainly included *tet*W-01, *tet*Q, *bla*SFO, *amp*C-02, *str*B, *aac*C, *sul*2, *opr*D, *pic*a, *mex*F, and so on while the ARGs clustered around the AL group (relatively higher body weight) mainly included *erm*B, *erm*F, *cfx*A, *mcr*1, *pen*A, *van*YD-01, *tet*(32), and so on. It seemed that the behaviors of ARGs were not simply elevated or decreased with the variation of their hosts’ body weight but rather there was a relatively complex process.

The relative abundance of MGEs was lower in mouse feces of the CR groups than that of the AL groups, especially the most abundant *tnp*A-03 and *tnp*A-06 genes, which decreased by 77% and 38%, respectively (Figure 2B). However, these two genes were the most increased MGEs in the feces of mice fed with high-fat diet [22]. In addition, IncQ plasmids, with the IncQoriT gene, increased obviously in the mouse feces of the CR groups, with abundance 8.8 times higher than that of the AL groups. In our previous study, the IncQoriT gene significantly decreased in the feces of mice fed with a high-fat diet [22]. All above data suggested that mainly MGEs, *tnp*A-03, *tnp*A-06, and IncQoriT might also be body weight-associated genes in mouse gut.

The RDA between ARGs and MGEs (Figure 3A) showed that the main ARGs in the mouse feces of the CR group (*tet*W-01, *opr*D, *amp*C-02, and *bla*SFO) clustered around *tnp*A-02, IncQoriT, pNI105map-F, and ISCR1. The main ARGs in the mouse feces of the AL group (*erm*B, *erm*F, *cfx*A, and *van*YD-02) clustered around *tnp*A-03 and *int*l-1. In addition, *mcr*1 showed a significant positive correlation with *tnp*A-06 in the AL group.

The relative abundance of pathogenic bacterial genes (Figure 2C), including 22S rDNA and *uid*A, increased significantly in the mouse feces of CR groups compared to those of the AL groups. It seemed that 22S rDNA and *uid*A genes were not body weight-associated genes because they also increased in the high-fat diet induced obese mouse feces.

### 3.3. Effect of CR on Microbiota Community

The gut microbial community was determined to further figure out the potential mechanism between microbiota, ARGs, and MGEs. Firmicutes and Bacteroidetes were the most important phyla in both the CR and the AL groups, which occupied 61.25–72.85% and 6.35–12.15%, respectively (Figure 4A). The ratio of Firmicutes to Bacteroidetes (F/B) showed an obvious decrease in the CR group (5.04) compared with those in the AL group (11.46). The F/B value has been frequently shown to have a positive association with body weight and might become a biomarker indicating body weight [32]. Interestingly, Actinobacteria occupied much higher proportions in the CR group (lower body weight) (18%) than that in the AL group (higher body weight) (5%) (*p* < 0.05). According to the data from our previous study, Actinobacteria were much more abundant in the mouse feces of the control group (lower body weight) (11%) than that of the HFD group (higher body weight) (2%) [22], which suggested that the lower the body weight, the higher the abundance of Actinobacteria in the intestine of mice. In addition, Verrucomicrobia significantly decreased in the mouse feces of the CR group compared with those of the AL group. Studies have suggested that CR as a dietary intervention reshaped the gut microbiota composition, and further affected host metabolism [33]. The gut microbiota under CR are deprived of 40–60% of nutrients and thus will shift towards favoring bacteria that can more efficiently harvest energy and depleting organisms that are less efficient [34].

The Wilcoxon tests were performed to compare the abundance of gut microbiota in mouse feces between AL and CR groups (Figure 5). The results showed that the abundances of *Lactobacillus*, *Bifidobacterium*, *Corynebacterium*, *Prevotella*, and *Staphylococcus* in the gut of the CR group were much higher than those of the AL group while the abundances of *Faecalibaculum*, *Allobaculum*, and *Akkermansia* in the gut of the CR group were much lower than those of the AL group (Figure 5C). At the species level, *Lactobacillus_johnsonii*, *Bifidobacterium pseudolongum*, *Corynebacterium_glutamicum*, *Bacteroides_sp*._CAG:927, and *Lactobacillus_reuteri* were the predominant species in the feces of the CR group while *Faecalibaculum_rodentium*, *Allobaculum_stercoricanis*, *Firmicutes_bacterium*_M10-2, and *Akkermansia_muciniphila* were the predominant species in the feces of the AL group (Figure 5D).

It was notable that *Lactobacillus*, mainly including *Lactobacillus_johnsonii* and *Lactobacillus_reuteri*, increased extremely in the feces of the CR group compared to those of the AL group. However, in the previous study, the abundance of *Lactobacillus* also increased in the feces of mice fed with a high-fat diet [22]. Therefore, it seems that *Lactobacillus* was not a body weight-associated bacterium because its abundance increased in mouse feces of both the calorie restricted diet and high-fat diet groups. *Lactobacillus*, a Gram-positive bacterium, is included in lactic acid bacteria (LAB), which could regulate the physiological state and lipid metabolism of the host and are beneficial bacteria for the intestinal tract [35]. Another significantly increased species in the CR group was *Bifidobacterium*_*pseudolongum* (species), which increased from 1.17% (AL) to 6.77% (CR). In our previous study, the abundance of *Bifidobacterium* in the feces of mice fed with a high-fat diet was significantly lower than in those fed with a low-fat diet [22]. Combining these data, we might come to the conclusion that *Bifidobacterium* was a the body weight-associated bacterium in mouse gut. *Bifidobacterium* (Gram-positive) is a representative of beneficial bacteria and is also a type of LAB [36].

*Faecalibaculum* and *Akkermansia* were the dominant bacteria in the feces of the AL group, which occupied 18.88% and 6.97%, and were much higher in abundance than those in the CR group (7.64% and 0.44%). Interestingly, *Faecalibaculum* and *Akkermansia* were also the dominant bacteria in the feces of mice fed with the low-fat diet compared to those fed with a high-fat diet in our previous study [22]. Thus, *Faecalibaculum* and *Akkermansia* were not body-weight-associated bacteria in mouse gut.

### 3.4. Effect of CR on Abundance of Functional Genes

Kyoto Encyclopedia of Genes and Genomes (KEGG) analysis of differentially genes is shown in Figure 6, of which the most enriched metabolic pathways in mouse gut microbiota were biosynthesis of amino acids, ABC transporters, and carbon metabolism in both the AL group and the CR group. Compared with the AL group, the calorie restricted diet (CR group) remarkably up-regulated the abundance of genes involved in ABC transporters, quorum sensing (QS), aminoacyl-tRNA biosynthesis, ribosomes, and pyruvate metabolism and down-regulated the abundance of genes that mainly participated in biosynthesis of amino acids and homologous recombination in mouse gut microbiota. Additionally, the abundances of beta-lactam resistance genes (Appendix A) also increased evidently in the gut microbiota of the CR group, which were consistent with the results of ARGs in the mouse feces.

QS is a bacterial cell-to-cell communication process that regulates many traits and gene expression, including ARGs and the related genes that contribute to AMR development [37]. QS affects the transmission of ARGs, HGT via promoting membrane permeability, tactic movement, biofilm formation, and so on [18]. ATP-binding cassette (ABC) transporters form one of the largest protein families, which are implemented in various metabolic processes, including multidrug resistance [38]. The up-regulation of the abundance of these functional genes in mouse gut might have contributed to the increase in ARGs in the CR group.

### 3.5. Correlations of ARGs, MGEs, and Microbiota Community

The RDA between microbial community and the ARGs showed that the major ARGs in the feces of the CR group (*tet*W-01, *tet*Q, and *bla*SFO) clustered around *Lactobacillus*, *Enterorhabdus*, *Corynebacterium*, and *Prevotella*, which had a significant positive correlation with those dominant bacteria of the CR group (Figure 3B). The Spearman correlation analysis between gut microbiota and ARGs showed that there were significant positive correlations between several probiotics and ARGs, including *Akkermansia_muciniphila*, *Akkermansia_muciniphila*_CAG:154 and *cfx*A, *pen*A, *erm*B, *erm*F, *tet*32, *van*YD-01, *van*YD-02; *Bifidobacterium_animalis*, *Bifidobacterium_pseudolongum*, *Lactobacillus_gasseri*, *Lactobacillus_johnsonii*, *Lactobacillus_reuteri* and *aac*C, *str*B, *amp*C-02, *bla*SFO, *sul*2, *tet*Q, *tet*W-01, *van*C-03, *acr*F, *mex*F, *opr*D, *opr*J, *pic*a. In addition, *Actinobacteria-bacterium*-UC5.1-1B11 and *Bacteroides_sp*._CAG:927 also exhibited significant correlations with *aac*C, *str*B, *amp*C-02, *bla*SFO, *sul*2, *tet*Q, *tet*W-01, *van*C-03, *acr*F, *mex*F, *opr*D, *opr*J, *pic*a (Figure 7).

Normally, probiotics, such as LAB, are living non-pathogenic microorganisms, and when administered in sufficient amounts, can cause beneficial effects, such as enhanced growth rates, improved immune response, and increased resistance to harmful bacteria [39]. However, the previous study reported that they have the potential to spread AMR to harmful bacteria [36], which might be attributed to the widespread usage of probiotic bacteria in conjunction with or in close association with antibiotic use, or rather misuse. Probiotic bacteria are known to harbor intrinsic and MGEs that confer resistance to a wide variety of antibiotics [40].

The Spearman correlations between gut microbiota and MGEs were further analyzed and the results are shown in Appendix A. Most intestinal bacteria, such as *Akkermansia_muciniphila*, *Akkermansia_muciniphila*_CAG:154, *Allobaculum_stercoricanis*, and *Firmicutes_bacterium*_M10-2 exhibited significant positive correlations with *tnp*A-03 (Appendix A). *Lactobacillus_johnsonii*, *Lactobacillus_reuteri*, *Lactobacillus_taiwanensis*, and *Lactobacillus_gasseri* had significant positive correlations with *tnp*A-02 and pNI105map_F. Interestingly, most LAB, including *Lactobacillus* and *Bifidobacterium*, showed remarkable negative correlations with *tnp*A-03. There was no significant positive correlation between the dominant *Faecalibaculum_rodentium* in the intestine of the AL group and any detected MGEs, implying that it did not carry mobile ARGs. All the above data suggested that most probiotic bacteria might carry mobile ARGs in close association with MGEs, which might contribute to the elevated abundance of ARGs in the feces of the CR group. Rokon-Uz-Zaman et al. showed the presence of tetracycline and beta-lactamase ARGs in some isolates of *Lactobacillus* spp. from commercial poultry probiotic products [35]. These ARGs could be transferred to E.coli through horizontal transfer, which is thought to present the highest risk of the spread of ARGs among animals, humans, and the environment [35].

### 3.6. Correlations of ARGs, MGEs, and Functional Genes

The abundance of genes involved in ABC transporters, biosynthesis of amino acids, lysine biosynthesis, ribosomes, valine, leucine, and isoleucine biosynthesis, and beta-lactam resistance had remarkable positive correlations with the main ARGs in mouse gut microbiota (Appendix A). It was evident that the most functional genes involved in biosynthesis of amino acids exhibited significant correlations with the main ARGs, but they were down-regulated in intestines of the CR group (Figure 6). Quorum sensing did not show a significant correlation with any ARGs in mouse intestines, although the abundances of relevant genes increased overall in the CR group(Figure 6). The abundance of genes involved in ABC transporters showed significant positive correlation with *str*B, *amp*C-02, *bla*SFO, *sul*2, *tet*Q, *tet*W-01, *van*C-03, *acr*F, *mex*F, *opr*J, and *pic*a, and they were up-regulated evidently in the intestines of the CR group (Figure 6), which suggested ABC transporter function might be related to ARGs or AMR development.

The Spearman correlations between functional genes and MGEs in mouse gut were further analyzed (Appendix A). The results showed that the abundance of functional genes was rarely significantly positively correlated with MGEs but exhibited significant negative correlations, such as for ABC transporters, ribosomes, lysine biosynthesis, and *tnp*A-03. However, functional gene abundances of biosynthesis of amino acids in mouse gut showed a significant positive correlation with *tnp*A-03. In addition, the abundance of beta-lactam resistance genes showed significant positive correlations with IncQoriT, IncNrep, pNI105map_F, and ISCR1, which indicated that most beta-lactam ARGs in mouse gut might be mobile ARGs.

## 4. Conclusions

The study first demonstrated the effect of a calorie restricted diet on the ARGs and MGEs in mouse feces and explored their relationship with gut microbiota and their functions. The results showed that the abundance of the total ARGs in feces of the CR group increased, especially tetracycline ARGs (*tet*W-01). The abundance of the MLSB ARGs (*erm*B and *cfx*A) decreased evidently in feces of the CR group. In addition, the total abundance of MGEs decreased evidently in the CR group, especially *tnp*A-03. In the meantime, the abundance of *Lactobacillus*, *Bifidobacterium*, *Corynebacterium*, and *Prevotella* in feces of the CR group increased obviously while the abundance of *Faecalibaculum*, *Allobaculum*, and *Akkermansia*in mouse feces of the CR group decreased remarkably. Combined with the previous study, we have concluded that several ARGs (such as *erm*B, *cfx*A, and *sul*2) and MGEs (*tnp*A-03, *tnp*A-06, and IncQoriT) could be defined as body weight-associated ARGs and related to body weight-associated bacteria, such as *Bifidobacterium*. The Spearman correlation analysis between gut microbiota and ARGs showed that several probiotics, such as LAB, were significantly positively correlated with ARGs (*tet*W-01), which might be the main contribution to the increase in ARGs of the CR group.

Based on these preliminary results, a large-scale population study will be carried out to validate these conclusions in the future. Moreover, in addition to restricting the feeding amount, controlling the feeding period has also been researched recently, e.g., intermittent feeding [41]. The effects of different dietary restriction regimens on the spread of ARGs will be considered to explore the influencing factors and potential mechanisms in the future.

## Figures and Tables

**Figure 1 nutrients-15-03198-f001:**
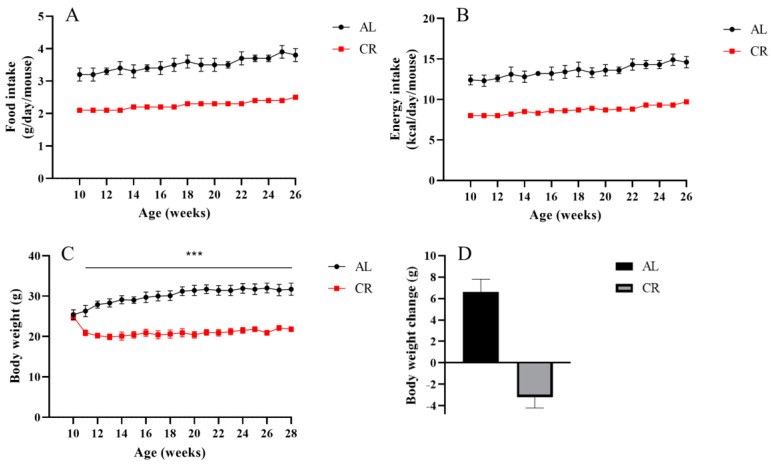
Effects of CR on body weight change in mice. (**A**) Food intake, (**B**) energy intake, and (**C**) body weight were measured weekly throughout the experimental period. (**D**) The average body weight change at the end of experiments. n = 10 per group. Data are shown as the means ± SD. *** compared to the AL group, *p* < 0.001.

**Figure 2 nutrients-15-03198-f002:**
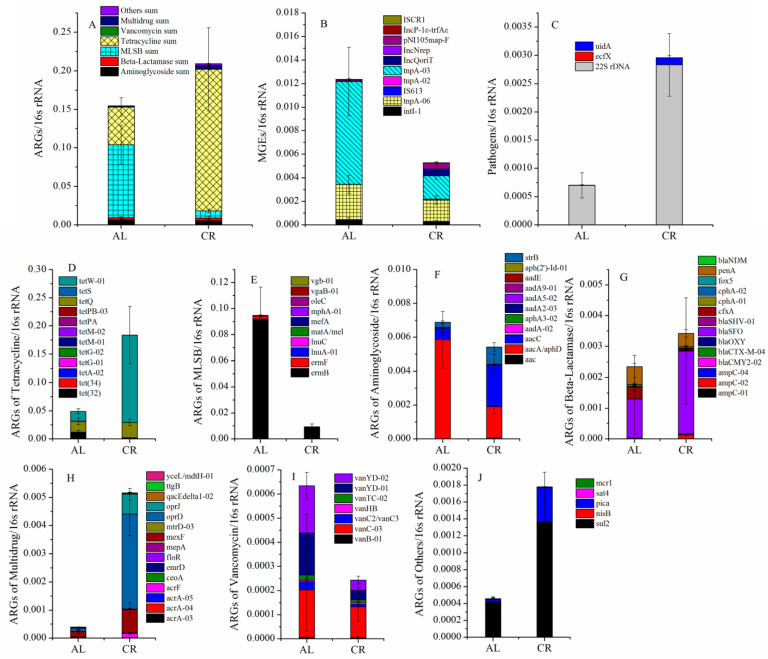
Effects of CR on the variations of the major ARGs in mouse feces. (**A**) ARG sum; (**B**) MGEs; (**C**) pathogens; (**D**) ARGs of tetracycline; (**E**) ARGs of MLSB; (**F**) ARGs of aminoglycoside; (**G**) ARGs of beta-lactamase; (**H**) ARGs of multidrug; (**I**) ARGs of vancomycin; (**J**) ARGs of others.

**Figure 3 nutrients-15-03198-f003:**
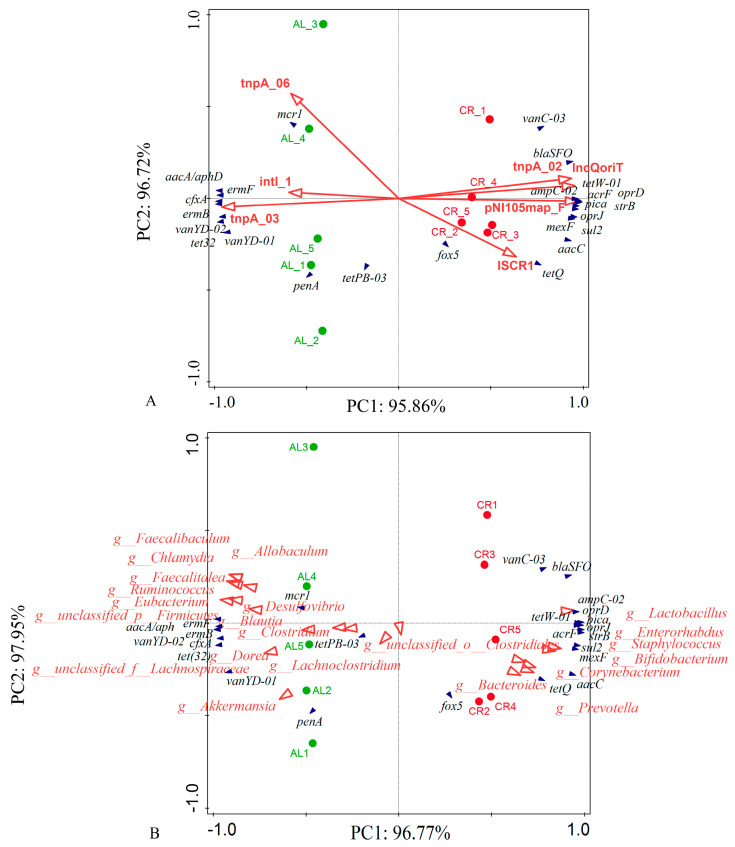
(**A**) RDA between ARGs (deep blue arrows) and MGEs (red arrows) in mouse feces; (**B**) RDA between ARGs (deep blue arrows) and gut microbiota (red arrows) in mouse feces (at genus level). The percentage of variation explained by each axis is showed, and the relationship was determined significant (*p* < 0.001) base on 499 permutations.

**Figure 4 nutrients-15-03198-f004:**
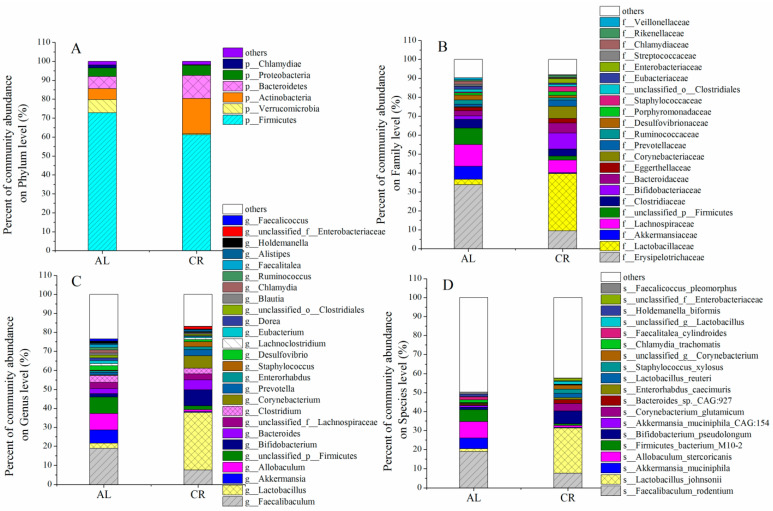
Effects of CR on gut microbiota in mouse feces. (**A**) On phylum level; (**B**) on family level; (**C**) on genus level; (**D**) on species level.

**Figure 5 nutrients-15-03198-f005:**
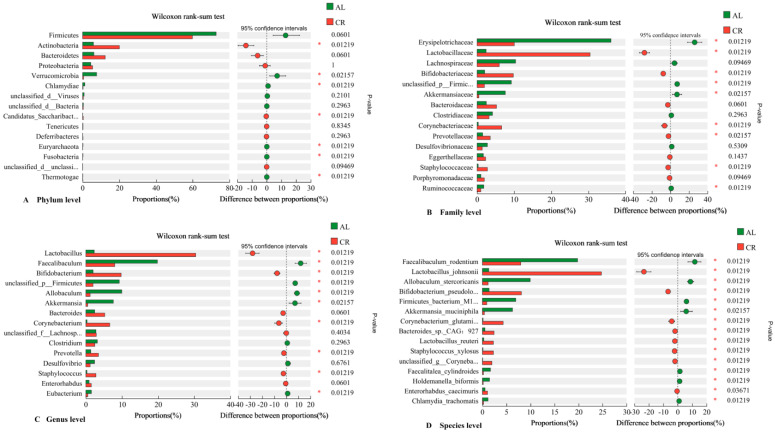
Wilcoxon rank-sum test of gut microbiota in mouse feces between AL and CR groups (*: *p* < 0.05).((**A**) Candidatus_Saccharibact…: Candidatus_Saccharibacteria; unclassified_d__unclassi…: unclassified_d__unclassified; (**B**) *unclassified_p__Firmic…: unclassified_p__Firmicutes*; (**C**) *unclassified_f__Lachnosp…: unclassified_f__Lachnospiraceae*; (**D**) *Bifidobacterium_pseudolo…: Bifidobacterium_pseudolongum; Firmicutes_bacterium_*M1*…: Firmicutes_bacterium_*M10-2; *Corynebacterium_glutami…: Corynebacterium_glutamicum;unclassified_g__Coryneba…:unclassified_g__Corynebacterium*).

**Figure 6 nutrients-15-03198-f006:**
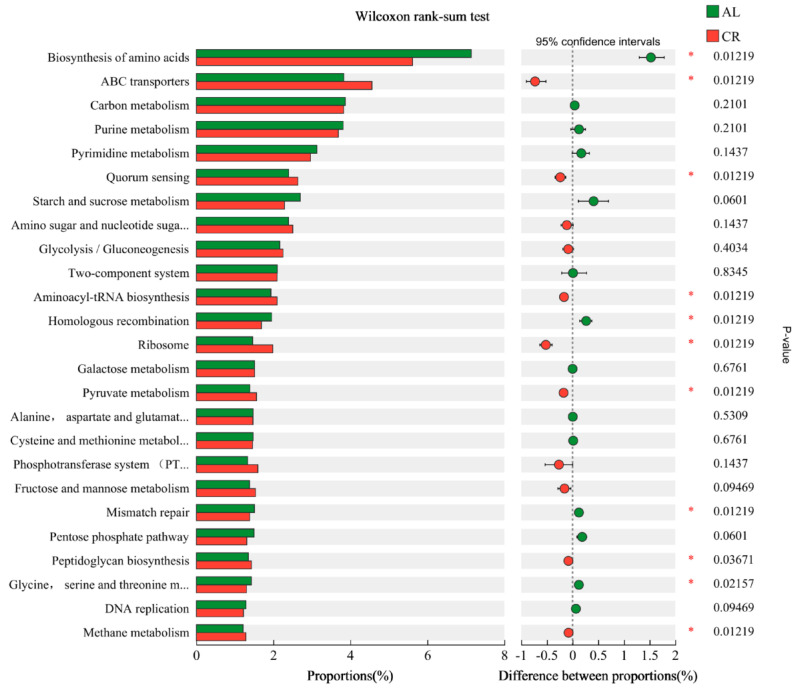
Wilcoxon rank-sum test of abundance of functional genes in mouse feces between AL and CR groups (*: *p* < 0.05). (Amino sugar and nucleotide suga…: Amino sugar and nucleotide sugar metabolism; Alanine, aspartate and glutamat…: Alanine, aspartate, and glutamate metabolism; Cysteine and methionine metabol…: Cysteine and methionine metabolism; Phosphotransferase system (PT…: Phosphotransferase system (PTS); Glycine, serine and threonine m…: Glycine, serine, and threonine metabolism).

**Figure 7 nutrients-15-03198-f007:**
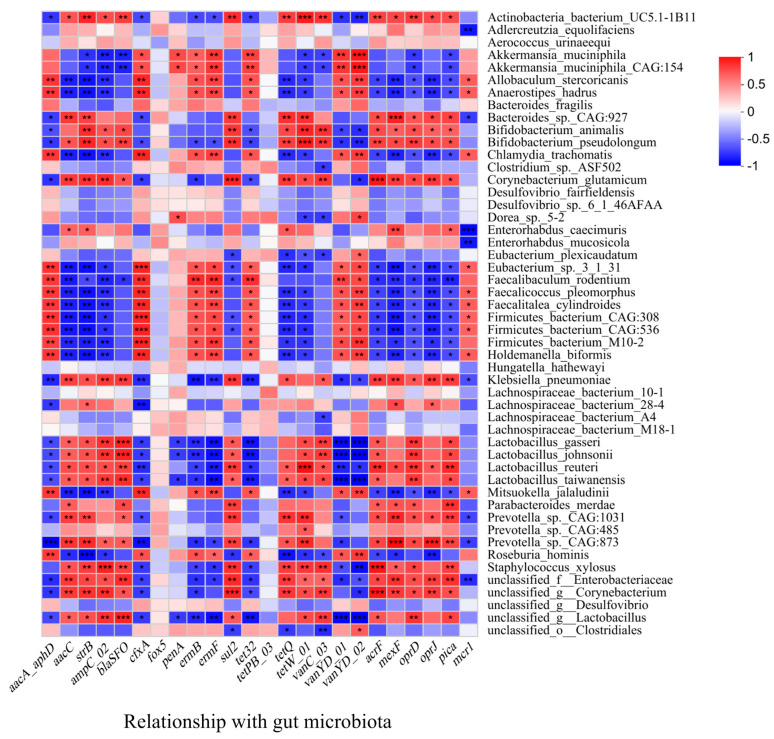
The Spearman correlations between gut microbiota and ARGs (at species level) (*: *p* < 0.05; **: *p* < 0.01; ***: *p* < 0.001).

## Data Availability

The original contributions presented in the study are included in the article. Further inquiries can be directed to the corresponding author.

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
