# Peer review of "An Insight into the Exploration of Antibiotic Resistance Genes in Calorie Restricted Diet Fed Mice"

_nutrients, 2023, doi:10.3390/nu15143198_

Round 1

Reviewer 1 Report

The manuscipt is well written and well presented

The authors should expand the statistical analysis section, to provide more details on the tests used. Sample size restrictions are evident as the p values are common among many tests and an explanation should be added on how that was dealt with. This should also be added in a limitation section.

The manuscript is well written and well understood

Reviewer 2 Report

I found this article very interesting and a very good continuation of the previous research contained in reference 11.
I would have a question: in the material and method part it specifies that the mice in the CR group were kept in individual cages. Were the ones in the AL group kept in the same cage? All 10 of them?
Also, the results , very interesting, are based on a single study group of only 10 mice. Perhaps, as a suggestion, it could be mentioned in the conclusions that these are preliminary results, which could possibly be developed further, on larger groups or possibly with a different dietary restriction regimen.

Reviewer 3 Report

This manuscript describes a unique challenge to evaluate the effects of calorie restriction on the amount of antibiotic resistance genes and gut microbiome. There are some questions about the contents.

1. Section 3.4 describes expression of functional genes in mouse feces. Material and Methods only explains extraction of DNA, but not mRNA. More explanation is necessary why gene expression could be analyzed.

2. Some discussion on the following questions would make the paper more interesting.

Why does microbiome change in response to calorie restriction?

Whether does each species tend to hold certain types of ARG? If so, why?

Other comments:

References

Line 35: Ref. 3 describes the trends of asbestos hazards and does not seem to authorize the statement in the text.

Line 37: Ref. 5 & 6 do not seem to describe the WHO’s suggestions directly.

Line 47: Ref. 11 does not seem to generally describe conjugation, transformation and transduction in HGT.

Line 49: Ref. 12 & 13 do not seem to evaluate the MGEs’ specific structure and ability to capture genes by site-specific recombination system.

Line 63: Reference is necessary after “the absence of exogenous chemical selective pressures.”

Line 323: The statement “The F/B value has been frequently shown a positive association…” describes a generally recognized fact. Therefore, a reference is necessary to support the statement.

Line 325: The statement “According to the data from our previous study, …” describes a kind of facts and, therefore, needs a reference. If “the data” have not been published yet, “(unpublished data)” should be added.

Line 384: The statement “In our previous study, …” needs a reference.

Line 393 The statement “in our previous study” needs a reference.

Line 486: Ref. 32 does not seem to support the text directly.

Figures

Figure 2, 4, 5 and 6: Letters in the figures may not look clear.

Some minor comments.

Line 69: Please add “(HFD)” after “high-fat diet” so that “HFD” can be used in the next paragraph (Line 80~).

Line 70: Please add “(CR)” after “calorie restricted.” 

Line 71: “aspectrum” would be “a spectrum”?

Line 74: “up to 50%” would be “up to 150%” or “up to by 50%.”

Line 100: Please add “(AL)” after “ad libitum” so that “AL” can be used in Line 103.

Line 119: “HT-qPCR” would be “high-throughput (HT)-qPCR.”

Line 153: “the redundancy analysis (RDA) analysis” would be just “the redundancy analysis (RDA).”

Line 211: “The RDA analysis” would be just “The RDA.

Line 381: “composed of” should be “included in” or “belonging to.”

Line 388: “it” should be deleted.

Line 421 & 468: Both “Functional gene expressions” and “functional genes expressions” may be “expression of functional genes.”

Line 430: “beta-Lactam resistance genes” are not involved in Figure 6.

Line 473: “, clustered” should be just “clustered.”

Line 484: “, when” should be “, and when.”

Line 488: “attribute” should be “be attributed.”

Line 549: “they down regulated …” needs evidence, which probably should be shown as “(Fig. 6).”

Line 552 and 555: Same as Line 549.
